# Interaction of Human Respiratory Syncytial Virus (HRSV) Matrix Protein with Resveratrol Shows Antiviral Effect

**DOI:** 10.3390/ijms252312790

**Published:** 2024-11-28

**Authors:** Thaina Rodrigues, Jefferson de Souza Busso, Raphael Vinicius Rodrigues Dias, Isabella Ottenio Lourenço, Jessica Maróstica de Sa, Sidney Jurado de Carvalho, Icaro Putinhon Caruso, Fatima Pereira de Souza, Marcelo Andres Fossey

**Affiliations:** 1Department of Physics, Institute of Biosciences, Humanities and Exact Sciences, São Paulo State University (UNESP), Rua Cristóvão Colombo, 2265, São José do Rio Preto 15054-000, SP, Brazil; thaina.rodrigues@unesp.br (T.R.); jefferson.busso@unesp.br (J.d.S.B.); rvr.dias@unesp.br (R.V.R.D.); isabella.otenio@unesp.br (I.O.L.); jessica.marostica@unesp.br (J.M.d.S.); sidney.carvalho@unesp.br (S.J.d.C.); icaro.caruso@unesp.br (I.P.C.); fatima.p.souza@unesp.br (F.P.d.S.); 2Multiuser Center for Biomolecular Innovation (CMIB), São Paulo State University (UNESP), São José do Rio Preto 15054-000, SP, Brazil

**Keywords:** respiratory syncytial virus, M protein, and resveratrol

## Abstract

The respiratory syncytial virus (RSV) matrix protein plays key roles in the virus life cycle and is essential for budding, as it stimulates the optimal membrane curvature necessary for the emergence of viral particles. Resveratrol, a polyphenol (3,4′,5-trihydroxy-trans-stilbene) produced by plants, exhibits pharmacological effects, including anti-inflammatory and antiviral activities. In this study, resveratrol was tested in HEp-2 (Epidermoid carcinoma of the larynx cell) cells for its post-infection effects, and recombinant M protein was produced to characterize the biophysical mechanisms underlying this interaction. The CC50 (Cytotoxic concentration 50%) value for resveratrol was determined to be 297 μM over 48 h, and the results from the HEp-2 cell cultures demonstrated a viral inhibition of 42.7% in the presence of resveratrol, with an EC50 (Half maximal effective concentration) of 44.26 μM. This mechanism may occur through interaction with the M protein responsible for the budding of mature viral particles. Biophysical assays enabled us to characterize the interaction of the M/resveratrol complex as an entropically driven bond, guided by hydrophobic interactions at the dimerization interface of the M protein, which is essential for the stabilization and formation of the oligomers necessary for viral budding. These findings suggest that one of the targets for resveratrol binding is the M protein, indicating a potential site for blocking the progression of the infection.

## 1. Introduction

Respiratory syncytial virus (RSV) is a prevalent cause of lower respiratory tract disease in infants and contributes significantly to morbidity and mortality among the elderly and immunocompromised adults [1]. It is estimated that 1 out of every 50 deaths in children younger than 5 years is attributable to RSV, indicating the virus’s substantial impact on global child health [2]. Although the majority of fatal cases occur among children residing in low- and middle-income countries, the significant short-term morbidity and economic repercussions result in a comparably high burden within high-income countries. In 2019, it was estimated that of the 101,400 global deaths attributed to RSV, 97% occurred in low- and middle-income countries [2].

Among the most commonly used therapies for respiratory syncytial virus (RSV) are ribavirin and the monoclonal antibody palivizumab. Both face challenges such as high costs, difficulties in maintaining dosage regimens, and severe adverse effects associated with their administration. Although new therapeutic proposals have been developed, they have thus far shown limited clinical success [3,4,5,6,7].

A recent study published in 2023 demonstrated an efficacy of 74.5% for the novel vaccine candidate nirsevimab in preventing RSV in healthy late-preterm and full-term infants [8]. However, to date, there are no specific antiviral therapies available for RSV infection.

Respiratory syncytial virus (RSV) is an enveloped virus that belongs to the *Orthopneumovirus* genus within the *Pneumoviridae* family [9,10]. Its non-segmented, negative-sense RNA genome encodes nine structural proteins, which include the envelope glycoproteins (F, G, and SH), the nucleocapsid proteins (N, P, and L), the nucleocapsid-associated proteins (M2-1 and M2-2), as well as the matrix (M) protein and two non-structural proteins (NS1 and NS2) [11,12].

The literature describes several classes of molecules with antiviral functions, particularly those targeting respiratory viruses, that hold therapeutic potential. Among these are polyphenolic phytoalexins such as resveratrol [13,14,15,16,17,18]. Resveratrol, a polyphenol (3,4′,5-trihydroxy-trans-stilbene) and a natural phytoalexin produced by plants, exhibits various pharmacological effects, including anti-inflammatory action, regulation of lipid metabolism, and antiviral properties. It is available in the trans- and cis-isomer forms; however, the cis-resveratrol isomer is unstable and easily transformed into the trans-form when reacted with light. It is insoluble in water but soluble in polar solvents such as ethanol and dimethyl sulfoxide [19].

The respiratory syncytial virus (RSV) matrix (M) protein plays a key role in the virus life cycle. It is initially localized in the nucleus during early infection, where it exerts an inhibitory transcriptional role. Subsequently, it accumulates in viral inclusion bodies before coordinating viral assembly and budding at the plasma membrane. The interaction between the M protein and cellular receptors involves proteins related to host transcription regulation, the innate immune response, cytoskeleton regulation, membrane remodeling, and cellular trafficking [20].

The viral M protein migrates from the nucleus to the cytoplasm through the nuclear transport protein CRM-1, interacting with a zinc finger region of the M2-1 protein. Together, they enter inclusion bodies, where the RNA nucleoprotein (RNP) complexes are present inside the viral particle [7]. At this stage, viral transcription terminates and packaging begins, with the M protein initiating the coordination and transfer of the RNP complexes located inside inclusion bodies to lipid raft regions, where the release of new virions occurs [18]. Therefore, the M protein is essential for budding, as it stimulates the optimal membrane curvature for the emergence of viral particles [20].

Structurally, the M protein comprises 256 amino acid residues and two distinct domains: the N-terminal and C-terminal domains, connected by a 13-amino-acid loop. The dimer interface consists of residues Ser63 to Pro68, Ala92 to Asp105, Val129 to Met134, Ile144, Tyr163, and Val225 to Ser235. The N-terminal domain has a horseshoe shape, while the C-terminal domain has a flattened barrel configuration, with both predominantly comprising β-sheets and an apparent binding site for divalent metals. Studies have demonstrated that the M protein can interact with itself and undergo self-oligomerization, facilitating the formation of oligomers through the interaction between the N-terminal region of one protein and the C-terminal region of another [20,21]. The positive residual charge on the M protein domains may facilitate its interaction with the negatively charged plasma membrane of pneumocytes, exposing certain hydrophobic regions for interaction with host proteins and between M protein monomers.

In this study, we tested resveratrol in HEp-2 cells for its post-infection effects. Additionally, we produced the M protein and characterized the parameters of the interaction between resveratrol and the M protein to elucidate the energy required for binding and how the hydrophobic character of the interaction appears to direct resveratrol to the protein dimerization interface.

## 2. Results

### 2.1. Effects of Resveratrol Cytotoxicity by MTT in HEP-2

For resveratrol, the cytotoxic concentration (CC50) was determined using the MTT assay at 24, 48, and 72 h (Appendix A). The values for each time point and concentration evaluated were 293 μM, 297 μM, and 225 μM, respectively. The average R-squared (R²) value for the analyzed data was 0.95, indicating the low toxicity of resveratrol at concentrations deemed safe for antiviral trials.

The CC50 of resveratrol in HEp-2 cells was found to be 297 μM at 48 h, demonstrating good levels of viability when compared to other phytoalexins [22]. When HEp-2 cells were infected with RSV at a multiplicity of infection (MOI) of 0.01 in the presence or absence of resveratrol, the results indicated that resveratrol performs effectively during the post-infection stage. The optimal tested concentration of 32 μM exhibited a significant reduction in viral replication compared to the control (Figure 1).

### 2.2. Biophysical Characterization of the M Protein Interaction with Resveratrol

A significant level of human respiratory syncytial virus (hRSV) matrix (M) protein was expressed and purified using Ni-NTA affinity resin, followed by size-exclusion chromatography. The peak volume fractions were obtained in the range of 10–12 mL, yielding a protein of approximately 30 kDa (Appendix A). The far CD-UV spectrum of the M protein at 298 K (Appendix A) was analyzed, revealing a major contribution from secondary β-sheet structures (44%), 23% α-helix, and 33% random coil. Below, in Appendix A, the far CD-UV spectrum of the M protein (5.0 µM) with the addition of resveratrol (50 µM) is presented. Although the spectral profile of the M protein/resveratrol complex at 50 µM showed variation, no significant change in the secondary structure of the protein was observed, which remained at 44% β-sheet, 20% α-helix, and 36% random coil structures.

The results obtained from the STD-NMR technique were utilized to investigate the interaction between the M protein and resveratrol. The difference spectrum for the M protein/resveratrol interaction is shown in Figure 2, indicating the presence of resonances at 7.50, 7.10, 6.90, 6.80, and 6.66 ppm, corresponding to the protons of the aromatic A-ring (hydrogens: 2–6) and B-ring (hydrogens: 3–5) of resveratrol, respectively. The signals at 6.90 and 7.10 ppm correspond to the alpha carbons between the A-ring and B-ring. The STD-NMR analysis revealed that the A-ring is more deeply embedded within the binding site of the M protein, mediated by the high polarity of the two hydroxyl groups present in the A-ring, which stabilize the hydrogen bonds with the protein. In the case of the B-ring, which contains only one hydroxyl group, interaction with the protein is observed, likely occurring through the stabilization of these hydrogen bonds. In summary, the amphipathic character of resveratrol facilitates interactions with the protein, stabilized by hydrogen bonds formed through contact between the hydroxyl groups and the hydrophobic interactions with the carbon groups of the molecule.

The quenching of the fluorescence intensity of the M protein (Figure 3) occurs due to several factors, including excited-state reactions, molecular rearrangements, energy transfer, ground-state complexation, and collisional quenching [23]. The analysis of the M protein fluorescence quenching during titration with increasing amounts of resveratrol at different temperatures (detailed in the Appendix A) demonstrated that as the concentration of resveratrol increases in the protein microenvironment, the fluorescence intensity of the M protein significantly decreases.

The stoichiometry (n) and the dissociation constant (K_b_) were determined using the double-log model by analyzing the fluorescence quenching during the resveratrol titrations. The thermodynamic parameters are summarized in Table 1, indicating the constants obtained through the analysis of the interaction parameters.

In the M protein/resveratrol complex, both the enthalpic and entropic contributions are positive (ΔG° > 0 and ΔS° > 0), indicating that the interaction is primarily driven by entropy and dominated by hydrophobic effects. This significant entropic gain arises from the displacement of water molecules from the solvation layer as resveratrol approaches the binding site. The presence of numerous hydrophobic subunits in peptides and drugs highlights the importance of hydrophobic interactions for ligand recognition by receptors. In this context, the hydrophobic characteristics of resveratrol are likely crucial for its anchoring within the hydrophobic pocket of the M protein (Figure 4).

### 2.3. Molecular Docking and Molecular Dynamics of M Protein Resveratrol Interaction

A detailed search for potential interaction pockets (sites) between the M protein and resveratrol indicated possible regions to accommodate the ligand. Among all the analyzed sites, two of them exhibit physical characteristics (size, hydrophobicity and presence of tryptophan) suitable for binding with resveratrol without limitations, called site1 and site 2, as shown in Appendix A. Site 1 is divided into alpha and beta sites, because they are symmetrical sites at the dimeric interface. All the docking calculations were meticulously repeated, resulting in a total of 1000 final conformations. Our results indicate interactions at site 1, with no favorable interactions observed at site 2. In Appendix A, the two best anchorings, based on the Amber Score Interaction Energy, are presented. Interestingly, the result is distributed between the symmetrical sites 1α and 1β.

To complement and refine the results of the molecular docking, we used the two best conformations to perform molecular dynamics analysis. In Figure 5A, the most representative conformation from the 300 ns trajectory is shown, which was used for calculating the contacts presented in Figure 5B. In Figure 5C, the structural parameter RMSD indicates convergence in the structural stability of the protein and ligand at both sites after 60 ns of MD. At site 1α, resveratrol adopts a highly stable conformation, observed during the dynamics. In this principal conformation, residues Asp28, Ser100, and Lys101 from chain A, and Lys156 and Glu233 from chain B, participate in hydrogen bonding with the protein’s oxygen, while residues Arg99 from chain A, and Val158, Trp242, Leu230, and Ile160 from chain B, participate in hydrophobic contacts. As for site 1β, resveratrol also remains stable, albeit with a higher degree of freedom in its movements. Among the interacting residues, the notable ones include Tyr104, Pro68, Tyr229, Lys232, Glu231, Asn93, and Lys101, which were not present in the previously reported binding mode.

The differences in the dynamics of resveratrol between the 1-alpha and 1-beta sites can be attributed to the distinct binding modes observed during molecular docking presented in Appendix A. In the 1-alpha site, the initial structure reaches an energy minimum, resulting in a stable conformation. In contrast, the 1-beta site undergoes a conformational exchange between the two different conformations. The key residues common to both sites include Glu233, Trp242, Leu230, Val158, and Lys101. Glu233 and Lys101 are responsible for hydrogen bonding, while Trp242, Leu230, and Val158 contribute to the hydrophobic nature of the interaction. On average, these residues are the primary contributors to the interaction, considering the conformational dynamics of the ligands at both sites.

## 3. Discussion

The concentration of resveratrol capable of producing 50% of its maximum effect (EC50) was determined to be 44.26 μM, and the selectivity index (SI) was found to be 6.7, with a *p*-value of <0.05. These data suggest that resveratrol plays a significant antiviral role in the post-treatment stage against respiratory syncytial virus (RSV).

In infections caused by RSV, there is an increase in the inflammatory processes within the lung epithelium, leading patients to a state of acute respiratory distress [7,8,9,10,11], which increases mortality and can result in chronic complications. The antiviral mechanisms and effects of resveratrol have been widely studied in relation to various viruses, including the influenza virus, hepatitis C virus [24], respiratory syncytial virus [24], varicella-zoster virus [25], Epstein-Barr virus [21], herpes simplex virus [21], human immunodeficiency virus [24], African swine fever virus, enterovirus, human metapneumovirus, and duck enteritis virus, as well as in models of multiple sclerosis, which can be induced by viral infection. In almost all of these studies, RSV demonstrated a notable reduction of viral infection, with the exception of multiple sclerosis and hepatitis C, where the disease progression worsened following administration of resveratrol [24,25,26,27,28,29,30,31,32].

As previously described, the respiratory syncytial virus (RSV) matrix (M) protein is responsible for interacting with the host cell membrane cytoskeleton, as well as directing the contents present in the viral particle for budding—steps that occur after the viral replication period [19,20,21,22]. To promote virion assembly and budding, the M protein has been shown to recruit cellular factors, including mitochondrial proteins implicated in the mitochondria-mediated stress response and apoptosis [21]. Early in infection, the M protein is transported into the nucleus by the host nuclear transport protein importin β1 [20], serving a dual role in inhibiting host cell transcription and avoiding potential suppression of viral transcription in the cytoplasm by the M protein [19]. The effects on host transcription, including the expression of nuclear genes encoding mitochondrial components, are strongly dependent on the association of the M protein with the host chromatin; mutations in the RSV M protein that reduce the chromatin association severely impact the production of infectious RSV [26]. Thus, our work also elucidates how the interaction of resveratrol with the M protein could occur and the mechanisms behind these interactions.

The STD-NMR results provide crucial insights into the interaction between the M protein and resveratrol, highlighting specific binding profiles. The epitopes of interaction indicate that the aromatic rings of resveratrol, particularly the A-ring, are deeply embedded in the M protein’s binding site, stabilized by hydrogen bonds due to its hydroxyl groups. Despite having only one hydroxyl group, the B-ring maintains effective contact with the protein, suggesting it contributes to the overall interaction stability. Resveratrol’s amphipathic nature facilitates these interactions through both hydrogen bonding and hydrophobic contacts. This integrative binding, characterized by precise stereochemical interactions and a balance of hydrophilic and hydrophobic forces, underscores resveratrol’s potential as a strategic antiviral agent targeting the M protein with substantial specificity and stability.

The fluorescence suppression of the M protein, reduced by approximately 93% due to resveratrol, indicates strong binding interactions within the protein’s microenvironment. The observed shift to longer wavelengths suggests differential quenching of tryptophan residues. The M protein contains two tryptophan residues: Trp242, located near the dimerization interface in the C-terminal domain, and Trp35, located in the N-terminal domain. Since there is not a complete quenching of tryptophan fluorescence, it is likely that resveratrol binds to the M dimer in a specific manner, where one or more tryptophan residues are influenced by the presence of resveratrol. This interaction potentially perturbs only certain tryptophan residues, offering insight into the binding specificity and site of interaction of resveratrol with the M protein.

In the computational experiments guided by the previously highlighted experimental results, we characterized the M/resveratrol complex interaction as an entropically driven bond facilitated by hydrophobic interactions with each monomer of the M protein. Molecular dynamics simulations demonstrated the stability of this bond through root mean square deviation (RMSD) values ranging from 0.1 to 0.2 nm, with ligand stabilization throughout the dynamics (Figure 5C). Hydrophobic contacts with residues such as Arg99 with Ring A and Trp242 with Ring B corroborate experimental data that showed disturbances in the chemical environment of tryptophan in solution. The interaction site is also stabilized by the hydrogen bonds between resveratrol and the protein. We identified a promising and suitable site for accommodating resveratrol, allowing the ligand flexibility for specific fits within the site. This site stands out as the most suitable, being predominantly hydrophobic, located in an accessible region of the protein at the dimerization interface, and featuring a nearby tryptophan (W242) that interacts with the ligand. It allows a 1:1 ratio of one molecule per protein and minimizes the ligand’s exposure to the solvent since it is not a superficial cavity. The main interacting residues mapped by molecular dynamics, as shown in Figure 5B, corroborate the fluorescence spectroscopy data (Figure 4A), indicating a hydrophobic contribution to the binding of resveratrol, which disturbs the chemical environment of tryptophan in solution.

## 4. Materials and Methods

### 4.1. Cell Culture and Viral Stock

The human laryngeal carcinoma cell lines (HEp-2; Cell Bank of Rio de Janeiro, Rio de Janeiro, RJ, Brazil) were maintained in Dulbecco’s modified Eagle’s medium (DMEM; from Sigma-Aldrich^®^, St. Louis, MO, USA) supplemented with 5% (vol/vol) fetal bovine serum (FBS; Cultilab, Campinas, SP, Brazil) and 1% of penicillin/streptomycin (25 μg/mL; Invitrogen, Carlsbad, CA, USA) in a humidified 5% CO_2_ incubator at 37 °C. HEp-2 cells are derived from the human respiratory tract and are permissive to RSV infection in cell culture [33]. The stain Long of human Respiratory Syncytial Virus (hRSV-Long) was provided by Dr. Karina Alves de Toledo (UNESP, Assis, SP, Brazil). The stock originating from the cell culture was obtained by the inoculation of HEp-2 cells adhered in 25 cm^2^ flasks. After achieving the cytopathic effect, the cells were collected with a cell spreader and centrifuged at 500 RCF. The supernatant was stabilized at −80 °C in 10% trehalose solution [34]. The viral titer was determined via a plaque reduction assay in HEp-2 cells, as previously described [35,36].

### 4.2. Cytotoxicity Assay

The cytotoxicity of resveratrol (CAS 501-36-0, 3,4′,5-Tri-hidroxi-*trans*-estilbeno, 5-[(1*E*)-2-(4-hidroxifenil)etenil]-1,3-benzenodiol, from Sigma-Aldrich^®^) was evaluated in HEp-2 cells using an MTT assay (Thiazolyl blue tetrazolium bromide, Sigma-Aldrich^®^), as described by Mosmann, with some modifications [36]. Monolayers of HEp-2 cells were adhered in 96-well plates and received the compound diluted in serum-free medium, in the range of 64 to 512 µM. The plates were incubated at 37 °C with 5% CO_2_ for 24, 48 and 72 h. After incubation, the medium was replaced with MTT solution (1 mg/mL) for 60 min and solubilized with DMSO (dimethylsulfoxide, Sigma-Aldrich^®^) to measure the absorbance in a plate reader (BioTek ELx808, Winoosk, VT, USA 96-wells) at 562 nm. The cytotoxic concentration capable of reducing 50% of the cell viability (CC_50_) was estimated by the dose–response curve using the GraphPAD Prism 8.0.1 software [32].

### 4.3. Antiviral Assays2

The viral inhibition of resveratrol was measured in a post-infection protocol, as performed on HEp-2 cells via a plaque reduction assay. At a confluence of 80–90%, the cells were infected at MOI 0.01 for 60 min and treated with non-cytotoxic concentrations of resveratrol, via diluting in the culture medium, in three independents trials. Non-infected and infected non-treated controls were also run in parallel [37,38]. The monolayers were overlaid with 1% CMC (carboxymethylcellulose) and incubated at 37 °C with 5% CO_2_ for 5 days. After plaque formation, the cells were fixed with 10% formaldehyde and stained with 1% crystal violet [34,35]. The plaque-forming units (PFUs) were counted to determine the effective concentration in reducing 50% of the infection (EC_50_) and selectivity index (SI). The statistical analysis was calculated by GraphPAD Prism 8.0.1 software using an unpaired *t*-test, with *p* < 0.05 considered significant.

### 4.4. Expression and Purification of the M Protein

*E. coli* RIL DE3 bacteria were transformed by the thermal shock method with the recombinant plasmid pD441-NHT:M (ATUM, Newark, CA, USA) from the RSV A2 strain. A colony was chosen and grown in Luria–Bertani (LB) medium with 50 mg/mL Chloramphenicol and 34 mg/mL Kanamycin at 37 °C until it reached an absorbance of 0.6. The expression was induced at 28 °C with 400 µM isopropyl b-D-1-thiogalactopyranoside (IPTG) for 16 h. After induction, the cell suspension was centrifuged, lysed, and purified by affinity chromatography and size-exclusion chromatography (Appendix A). The sample purity and detection were assessed by 15% SDS-PAGE. The protein concentration was determined spectrophotometrically. The UV-Vis (Cary-3E, Varian, Palo Alto, CA, USA) at 280 nm using the molar extinction coefficient of 27.390 M^−1^ cm^−1^ was determined by the ProtParam tool Version 2.0.5 from webserver ExPASy. 

### 4.5. Secondary Structure Analysis of the Protein in the Presence and Absence of Resveratrol

The secondary structure analysis of the M protein was performed by the circular dichroism (CD) technique using a Jasco 815 spectropolarimeter (Jasco, Easton, MD, USA) equipped with a Peltier-type temperature control system and a 1.0 mm path-length quartz cuvette, which was coupled to a metal spacer block. The far UV-CD spectra of the M protein were recorded with 10 scans in the 260–190 nm range using a scan speed of 50 nm/min, response time of 1.0 s, spectral bandwidth of 1.0 nm, and spectral resolution of 0.2 nm. The buffer’s contribution to the CD spectra (50 mM Na_2_HPO_4_/NaH_2_PO_4_ (pH 6.5), 150 mM NaCl, 1.0 mM DTT) was subtracted from the protein spectra. The CD spectra were recorded as milligrams and then expressed in terms of the molar ellipticity (*Θ*) in units of deg·cm^2^·dmol^−1^. The percentage of secondary structures of the M protein was calculated with the CONTIN software (http://dichroweb.cryst.bbk.ac.uk/html/home.shtml) from the CDPro package, using the reference set of proteins SMP56 [39].

### 4.6. Saturation Transfer Difference NMR Spectroscopy

The STD-NMR experiments were recorded on a Bruker Avance III HD 600 MHz spectrometer equipped with a cryogenically cooled z-gradient probe (Bruker BioSpin GmbH, Rheinstetten, Germany). The 1H-NMR spectra were recorded on a Bruker Avance III 600 MHz spectrometer equipped with a 5 mm triple resonance cryoprobe and field gradient in the z-axis. All the data were analyzed with Bruker’s TopSpin program version 3.2. A 10 μM M protein solution (50 mM Na_2_HPO_4_/NaH_2_PO_4_ pH 6.5, 100 mM NaCl, 1 mM DTT) was used to determine the best saturation conditions, with frequencies close to 0 ppm and keeping the off-resonance frequency at 40 ppm. The saturation time was set to 2 s, with a recycle time of 3 s. A total of 128 scans were collected with 4 dummy scans and a saturation power of −35 dB. In the experiments with 150 μM resveratrol added, a 30 ms spin lock filter was used to suppress the signals coming from the M protein at 15 μM. The STD effect (ISTD) on a given proton of the ligand was calculated according to the following equation [40,41]:(1)$$I_{\mathrm{STD}}=\frac{I_{off\;resonance}-I_{on\;resonance}}{I_{off\;resonance}}$$
where I_on resonance_ and I_off resonance_ are the intensities of the ligand signals in the on- and off-resonance spectrum, respectively. The proton of the resveratrol with the highest STD effect (magnetization transfer maximum) was equal to 100% and the others protons were normalized according to this signal.

### 4.7. Analysis of the M Protein Interaction with Resveratrol Through Fluorescence Spectroscopy

The fluorescence spectroscopy measurements were conducted using an ISS PC1 spectrofluorometer (Champaign, IL, USA) equipped with a Neslab RTE-221 thermal bath (Thermo Electron Corporation, Waltham, MA, USA). A quartz cuvette with a one-centimeter optical path was employed. The excitation wavelength for the M protein tryptophans was set at 295 nm, and the wavelength range for the spectrum collection was from 300 nm to 500 nm. The measurements were carried out at different temperatures (288, 298, and 308 K) with an M protein concentration of 5.0 μM and a final concentration of 50 μM of resveratrol.

For the data analysis, it was hypothesized that the binding sites of the resveratrol on the M protein are equal and independent. Thus, the binding constant (Kb) and the number of binding sites (n) can be calculated using the following equation:(2)$$log\;\left(\frac{F_{0}-F}{F}\right)=n.log.K_{b}+n.log\;\frac{1}{\left(\left[LT\right]-\left(\frac{F_{0}-F}{F_{0}}\right)\left[M\right])\right)}$$
where F_0_ and F are the fluorescence intensity of the M protein in the absence and in the presence of the resveratrol, respectively. [M] and [LT] are the total concentration of M protein and ligand (resveratrol), respectively. The values of K_b_ and *n* for the M/resveratrol complex are obtained from the ordinate and slope of the linear fitting from the double-log plot log ((F_0_ − F)/F) as a function log(1/([LT] − ((F_0_ − F)/F_0_)[PT])), respectively.

The Van’t Hoff analysis allows us to better understand each contribution, providing the thermodynamic profile of the interaction. The standard Gibbs free energy variation is related to the association constant Kb through the following relationship:(3)$$\Delta G^{0}=-RTln(K_{b})$$
where R = 1.98 cal mol^−1^ K^−1^ 1 is the universal gas constant and K_b_ is the association constant referring to the temperature used in the experiments. The variation of K_b_ with the temperature T is given by the Van’t Hoff equation:(4)$$\frac{dRln(K_{b})}{d\left(\frac{1}{T}\right)}=-{\Delta H}^{0}$$
where ${\Delta H}^{0}$ is the standard enthalpy variation that refers to the internal energy variation of molecular species in solution caused by the binding process. Therefore, the slope of the curve of $-Rln(K_{b})$ as a function of (1/T) provides the value of ${\Delta H}^{0}$, and the value of the standard entropy variation ${\Delta S}^{0}$ at each temperature can be obtained by $\Delta G^{0}=\Delta H^{0}-T\Delta S^{0}$ [31].

### 4.8. Molecular Docking

Molecular docking simulations of the M protein/resveratrol complex were conducted using the three-dimensional information obtained from the Protein Data Bank [42] (PDB ID: 4V23), alongside the three-dimensional structure of resveratrol identified by its compound identification number (CID: 445154). Molecular docking calculations were performed using the UCSF Dock 6.7 package [43]. This software allows us to use the UCSF Chimera version 1.18 [44] program in conjunction with it for ligand and receptor preparation, as well as for visualization of the results. The grid box utilized was constructed by extending 8 Å from the center of each cluster in the x, y and z directions. This expanded box is then subdivided into bins of 0.2 Å, with a distance tolerance of 0.75 Å established for matching ligand atoms to the receptor. The initial stage of molecular docking involved the use of the Single Grid Energy (SGE) score function, which includes the electrostatic and van der Waals interaction components. In this calculation, the protein remains fixed while the flexible ligand is adjusted using a method known as the “anchor and grow” algorithm. During each docking run, 1000 ligand orientations are tested, and only the one with the lowest energy is selected. This entire procedure is repeated 1000 times, resulting in a total of 1000 final conformations, which enables a statistical analysis of the generated conformations. Additionally, a rescoring ranking was applied to the conformations obtained from the SGE through the Amber Score Binding Energy (ASBE). The method uses Ecomplex = Ebinding − (Ereceptor − Eligand), where Ecomplex, Ereceptor, and Eligand are energies approximated by the Amber force field with the MM-GB/SA method [45] This second method is important since it allows degrees of flexibility in the pocket and ligand.

### 4.9. Molecular Dynamics

All the stages of the molecular dynamics analysis were conducted using the GROMACS software package, version 5.0.7 [46]. Parametrization of resveratrol was carried out using the ATB-Server version 3.0 [44,47], where the parameters were preformatted for use with the GROMOS 54A7 force field [46]. The parameters followed a standard protocol of replica and time steps, with a total of 1.5 × 10^7^ steps and a time step of 2 fs, resulting in a computational time of 300 ns for each replica. The temperature and pressure were kept constant using the V-rescale and Parrinello–Rahman algorithms, with values of 300 K and 1 atm, respectively. The systems were neutralized by replacing some water molecules with chloride (Cl^−^) and sodium (Na^+^) ions, resulting in a total concentration of 150 mM.

Preceding the dynamic simulations, minimization and equilibration steps were performed to remove steric constraints and prepare the system. To achieve this, 5 × 10^5^ energy minimization steps were carried out using the steepest-descent method with all the flexible atoms, followed by an additional 5 × 10^5^ energy minimization step using the conjugate gradient method. Subsequently, system equilibration was conducted in two stages. First, a 1000 ps dynamics run in the NVT ensemble with position restraints applied to the protein and ligand atoms, allowing the solvent to adjust appropriately. This was followed by another 1000 ps dynamics run in the NPT ensemble, also with position restraints, to ensure the system adapted to the thermodynamic conditions. MD analyses were performed using tools available in the GROMACS package [44].

## 5. Conclusions

This study concludes that resveratrol demonstrates low cytotoxicity in HEp-2 cell cultures and exhibits antiviral activity during the post-treatment phase, proving effective against respiratory syncytial virus (RSV). Furthermore, it interacts strongly with the virus’s M protein, primarily through hydrophobic interactions and hydrogen bonds at site 1 of the protein. Collectively, these findings suggest resveratrol’s potential as a therapeutic agent for inhibiting the spread of RSV. This study highlights how resveratrol can exert antiviral effects and its interaction with the M protein. Therefore, further investigations using animal models and various cell lines could help reaffirm its antiviral potential and elucidate the underlying mechanisms of action.

## Figures and Tables

**Figure 1 ijms-25-12790-f001:**
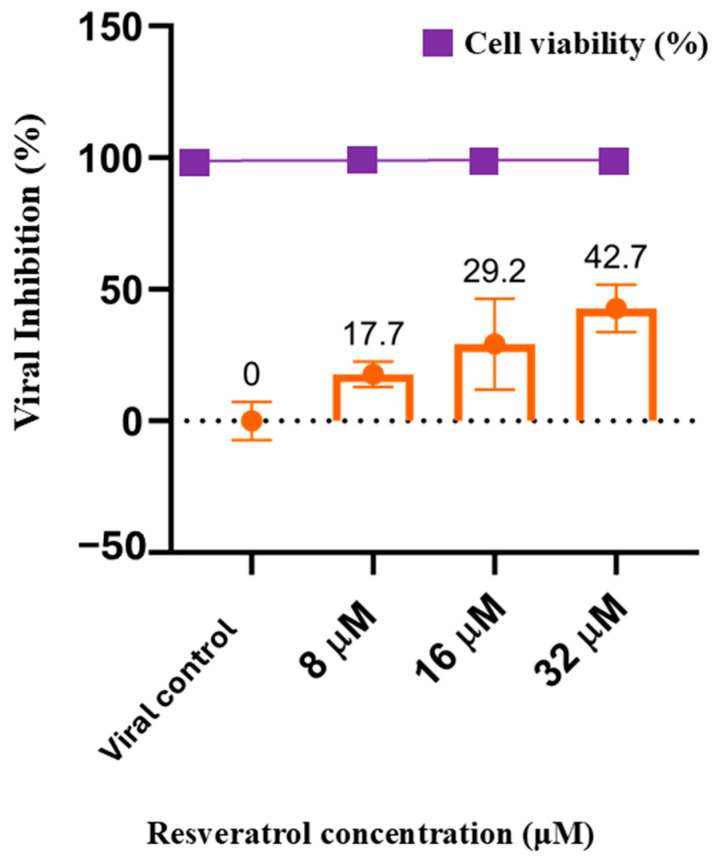
Percentage of viral inhibition in plaque formation during the post-treatment stage in the presence of different concentrations of resveratrol (colored in orange) and the cytotoxicity of resveratrol (purple) in the concentrations analyzed. On the x-axis, we have the viral control and the three different concentrations of resveratrol tested, and on the y-axis, the percentage of viral inhibition compared to untreated infected cells.

**Figure 2 ijms-25-12790-f002:**
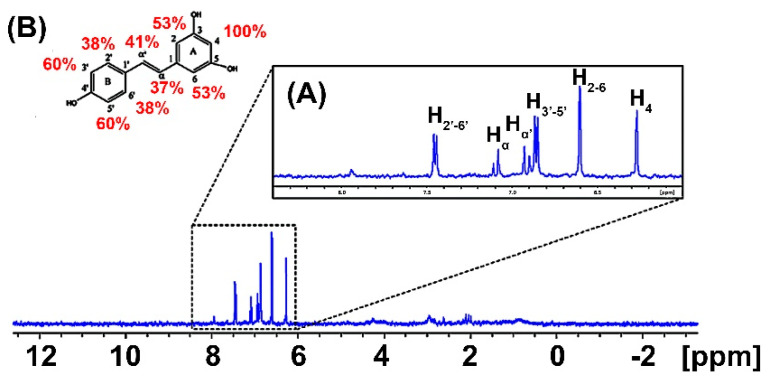
STD-NMR experiments on the M protein/resveratrol binding at 298 K. (**A**) The 1D 1H-NMR difference (blue) spectrum of complexes with M protein (15 µM)/resveratrol (200 µM). (**B**) Percentages of binding epitopes (red) indicated in the molecular structure of resveratrol near to the atom names.

**Figure 3 ijms-25-12790-f003:**
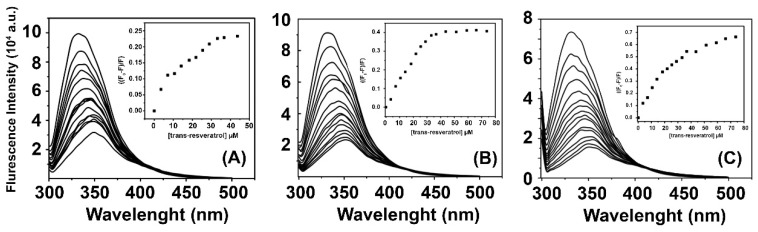
Normalized emission spectra of the fluorescence quenching of M protein by the resveratrol at (**A**) 288 (**B**) 298 (**C**), and 308 K with an excitation wavelength of 295 nm, from 0 to 50 μM titrated into 50 μM protein solution. The insert corresponds to the maximum emission wavelength of M protein in the presence of resveratrol and saturation curve.

**Figure 4 ijms-25-12790-f004:**
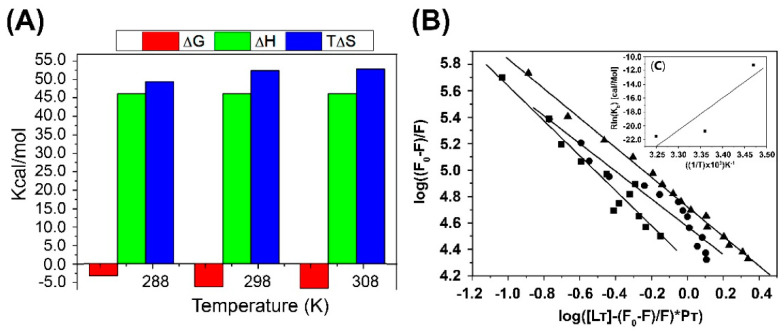
(**A**) Thermodynamic profile of the M protein/resveratrol complex at temperatures 288, 298 and 308 K. In red is the variation of the Gibbs free energy (ΔG), in green is the variation of the enthalpy (ΔH) and in blue is the entropy variation (TΔS). (**B**) The linearity behavior for the experiments performed at three different temperatures (288, 298 and 308, represented by square, circle and triangle, respectively) reveals that the interaction of resveratrol with the M protein agrees with the model, where the ligand interacts with the M protein. (**C**) The insert corresponds to the graph of the Van’t Hoff plot and thermodynamic analyzes of the M protein and resveratrol complexes at the three temperatures to characterize the type of interaction in solution.

**Figure 5 ijms-25-12790-f005:**
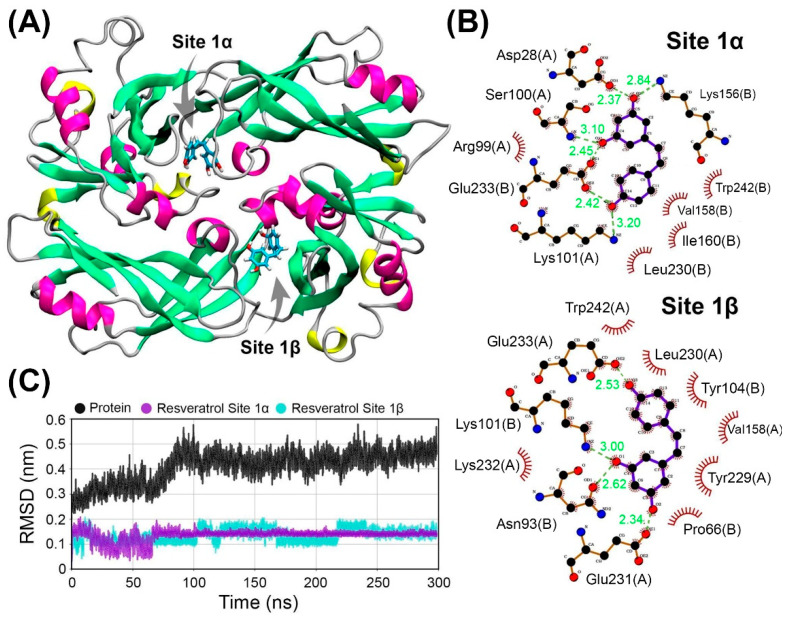
Molecular dynamics and interaction between the M protein and resveratrol. (**A**) The most representative structure of the molecular dynamics, indicating that there are degrees of flexibility for the ligand within the sites, allowing it to make small movements to accommodate itself in the protein. The M protein is represented in cartoon format and colored according to its secondary structures, with alpha helices in pink and yellow (3–10 helices), beta sheets in green, and random coils in gray. The ligand is colored in cyan, with its hydrogens in white and oxygens in red. (**B**) Molecular interactions of the most representative bindings. Both bindings are stable and share some common residues. The red tendrils indicate residues with hydrophobic interactions, and the green dashed lines represent hydrogen interactions. (**C**) RMSD of the molecular dynamics, separating the protein and resveratrol at sites 1α and 1β. After 70 ns of dynamics, all the structures converged in terms of their stability.

**Table 1 ijms-25-12790-t001:** Values referring to the M protein/trans-resveratrol protein complex at temperatures of 288, 298 and 308 K. Association constant K_b_, binding sites (n) and thermodynamic parameters at each temperature, ΔH, ΔG and ΔS.

T (K)	∆G (kcal/mol)	∆H (kcal/mol)	∆S (kcal/mol)	n	K_b_ (×10^4^ M^−1^)
288	−4.5 (±0.4)	26.5 (±0.1)	31.1 (±0.3)	0.9 (±0.05)	1.8 (±0.1)
298	−5.9 (±0.6)	26.5 (±0.1)	32.4 (±0.5)	0.9 (±0.06)	4.1 (±0.9)
308	−6.6 (±0.1)	26.5 (±0.1)	33.1 (±0.1)	0.9 (±0.01)	5.2 (±0.6)

## Data Availability

The data used to support the findings of this study are available from the corresponding authors upon request.

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
