# Peer review of "Interaction of Human Respiratory Syncytial Virus (HRSV) Matrix Protein with Resveratrol Shows Antiviral Effect"

_ijms, 2024, doi:10.3390/ijms252312790_

Round 1
Reviewer 1 Report
Comments and Suggestions for Authors
The current manuscript is describing the use of resveratrol as antiviral agent against Respiratory Syncytial Virus. the experiments are well designed and well thought. However, some questions were raised.
Having in mind that resveratrol is an isomeric molecule and that easily trans resveratrol can convert to cis resveratrol by light (UV). Does the authors need explain and say the two possibilities of resveratrol. Most likely it is trans-resveratrol, but the authors need to explain in the manuscript to eliminate doubts.
A little lack of scientific rigor was found the authors must talk about PMID: 26693226 and manuscript from there.
Please include the source and purity of resveratrol, I did not find it.
It is well-known that the two resveratrol's can have different pharmacological activities in biological systems (PMID: 33244652). based on that are the authors totally sure that the results are involving only trans resveratrol cis or both? How can they solve that?
The NMR experiments are a plus. but I have a question which is the best concentration to see the interaction. is not 50 uM excess and to concentrated?
Can you see which are the residues of the protein involved in the interaction also by NMR?
A paragraph of future studies (what is next) can help to have high value to the manuscript.
Overall, good manuscript but the authors need to correct the imprecisions in it.
Comments on the Quality of English LanguageMinor and for example do not use abbreviations in the abstract or use it and indicate.
Author Response
The current manuscript is describing the use of resveratrol as antiviral agent against Respiratory Syncytial Virus. the experiments are well designed and well thought. However, some questions were raised.
Having in mind that resveratrol is an isomeric molecule and that easily trans resveratrol can convert to cis resveratrol by light (UV). Does the authors need explain and say the two possibilities of resveratrol. Most likely it is trans-resveratrol, but the authors need to explain in the manuscript to eliminate doubts.
Thank you for your comment. In our work, we used trans-resveratrol CAS 501-36-0 - Resveratrol, ≥99% (HPLC) from Sigma-Aldrich.
A little lack of scientific rigor was found the authors must talk about PMID: 26693226 and manuscript from there. Please include the source and purity of resveratrol, I did not find it.
Thank you for your comment, as previously answered, the purity of resveratrol is 99%. Additionally, we included it in our work in section Materials and Methods 4.2 “Cytotoxicity Assay” the information on purity, origin, and isomeric trans.
It is well-known that the two resveratrol's can have different pharmacological activities in biological systems (PMID: 33244652). based on that are the authors totally sure that the results are involving only trans resveratrol cis or both? How can they solve that?
Thank you for your comment. In our study, we investigated the interaction of trans-resveratrol. It is important to note that the cis-resveratrol isomer may exhibit different effects in such interactions. However, our research focused solely on the isomeric trans, which is recognized for its greater stability compared to the cis form. The resveratrol stock was stored protected from light and maintained at the controlled temperatures specified by the manufacturer, and it was used only at the time of the experiment. To explore the interactions between the M protein and cis-resveratrol, further studies would be required. However, as previously stated, this was not the primary focus of our work.
The NMR experiments are a plus. but I have a question which is the best concentration to see the interaction. is not 50 uM excess and to concentrated? Can you see which are the residues of the protein involved in the interaction also by NMR?
Thank you for your comment. STD-NMR experiments are performed from the point of view of saturation of the ligand concentration relative to the protein. Therefore, the STD NMR technique is effective for systems with μM to mM affinity, and as such is ideally suited to studying the binding of ligand fragments. In our study, we employed only the STD-NMR technique, which analyzes the interaction from the perspective of the ligand to characterize the involved interaction epitopes. While experiments such as HSQC conducted in NMR can yield information about the protein residues that participate in the interaction, this experiment was not performed for the RSV M protein.
A paragraph of future studies (what is next) can help to have high value to the manuscript.
Thank you very much for your suggestion. We have added insights from our work to the conclusion to enrich the text and inform the reader about the new possibilities.
Comments on the Quality of English Language: Minor and for example do not use abbreviations in the abstract or use it and indicate.
Thank you for your observation. I have removed the abbreviation and highlighted the changes in the text in yellow.

Reviewer 2 Report
Comments and Suggestions for Authors
The study requires linguistic assessment because there are numerous grammatical problems. It must also demonstrate the significance of this study and the degree to which people with an interest in the subject matter depend on its findings. The reader may find a great deal of information in this research to be ambiguous or difficult to understand. In vitro research is also required to strengthen it further.
Comments on the Quality of English LanguageThe manuscript has a number of English language blunders, for example:
Line 21: recombinant M protein was produced the to characterize’.
Line 60: Several published works show its protection against the human respiratory syn-cytial virus and its role as an anti-inflammatory acting in the promote the decrease of cel-lular inflammation through the suppression of genes, decrease of cytokines and interleu-kins such as IL-6, TNf-α and other cellular modulators
Author Response
The study requires linguistic assessment because there are numerous grammatical problems. It must also demonstrate the significance of this study and the degree to which people with an interest in the subject matter depend on its findings. The reader may find a great deal of information in this research to be ambiguous or difficult to understand. In vitro research is also required to strengthen it further.
Thank you for your comment and suggestion. We have reviewed the vocabulary and grammar of the text. The initial in vitro experiments will inform and guide subsequent antiviral assays aimed at elucidating the interactions in cell culture.
Comments on the Quality of English Language
The manuscript has a number of English language blunders, for example:
Line 21: recombinant M protein was produced the to characterize’.
Line 60: Several published works show its protection against the human respiratory syncytial virus and its role as an anti-inflammatory acting in the promote the decrease of celular inflammation through the suppression of genes, decrease of cytokines and interleukins such as IL-6, TNf-α and other cellular modulators.
Thank you for the corrections; we have made the changes in the text to correct the errors.

Reviewer 3 Report
Comments and Suggestions for Authors
The Abstract is written synthetically and clearly, correctly, in accordance with the recommendations. Although the analytical methods are not indicated, the reader has an idea of ​​the analytical spectrum used in the experiment by presenting the results from each of them.
The introduction fully explains the needs and importance of the undertaken research and indicates innovative elements for the experiments performed.
The results were presented in a clear, transparent way. The graphs and figures were correctly described. The research results are interesting and valuable for the development of knowledge in the field of developing effective methods of protection against the RSV virus.
1. Suggestion: The only deficiency appears in the very short summary of the Conclusion section. I believe that it should be improved and the most important observations regarding the undertaken research should definitely be included, even if they are partly repeated with fragments of the Abstract.
I recommend the article for publication.
Author Response
The Abstract is written synthetically and clearly, correctly, in accordance with the recommendations. Although the analytical methods are not indicated, the reader has an idea of ​​the analytical spectrum used in the experiment by presenting the results from each of them. The introduction fully explains the needs and importance of the undertaken research and indicates innovative elements for the experiments performed. The results were presented in a clear, transparent way. The graphs and figures were correctly described. The research results are interesting and valuable for the development of knowledge in the field of developing effective methods of protection against the RSV virus.
Thank you for your comment and in-depth look at our article highlighting the important points of the reading.
Suggestion: The only deficiency appears in the very short summary of the Conclusion section. I believe that it should be improved and the most important observations regarding the undertaken research should definitely be included, even if they are partly repeated with fragments of the Abstract.
Thank you very much for your comments and suggestions. We have revised the conclusion to include additional information and enhance the text.

Reviewer 4 Report
Comments and Suggestions for Authors
In this manuscript, the authors investigate resveratrol, a polyphenolic compound, as a potential antiviral agent against respiratory syncytial virus (RSV), focusing on the inhibition of the viral M protein, which plays a crucial role in viral replication. Their study, conducted on HEP-2 cells, demonstrated that resveratrol effectively inhibits RSV replication post-infection with an EC50 value of 44.26 μM while exhibiting low toxicity.
To clarify the mechanism of action, the authors investigated the binding interaction between resveratrol and the viral M protein using STD-NMR and fluorescence spectroscopy. These experiments confirmed the binding of resveratrol to the M protein, suggesting that the interaction is mainly driven by hydrophobic effect. In addition, they identified a tryptophan residue which interacts with the ligand.
Building on these findings, the authors used molecular docking and molecular dynamics simulations to predict the binding site and investigate the interactions. The simulations showed that binding site 1 is the most likely interaction site. The study further showed that the interaction between resveratrol and the M protein is stabilized by a combination of hydrogen bonding and hydrophobic interactions. This site, located at the dimerization interface of the M protein, includes a tryptophan residue (W242). The methods used were well performed and adequately described.
However, I have a few key comments:
The authors mention the determination of EC50 and selectivity index (SI) in their methods for the antiviral assays, but these data are missing in the results section. Only inhibition values for three concentrations are given. It would be advantageous to provide the complete concentration-response data for which the EC50 was determined. This would give a clearer picture of the antiviral efficacy of the compound.
In the "Molecular Docking and Molecular Dynamics of M Protein-Resveratrol Interaction" section of the results, the authors state that two different ligand conformations were observed at binding site 1-beta, but only one conformation is shown. This discrepancy needs to be addressed, and the manuscript should include visualizations or discussions of both conformations.
The first three paragraphs of the Discussion focus on the inflammatory response associated with RSV infection. This content seems unnecessary, since the study does not directly examine inflammation or immune responses.
The manuscript lacks a comprehensive discussion of the toxicity and antiviral activity of resveratrol. This would be an important addition to better contextualize the therapeutic potential of resveratrol.
In line 290, the authors state that "Molecular dynamics simulations demonstrated the stability of this bond through RMSD…." but no RMSD values are provided.
Author Response
In this manuscript, the authors investigate resveratrol, a polyphenolic compound, as a potential antiviral agent against respiratory syncytial virus (RSV), focusing on the inhibition of the viral M protein, which plays a crucial role in viral replication. Their study, conducted on HEP-2 cells, demonstrated that resveratrol effectively inhibits RSV replication post-infection with an EC50 value of 44.26 μM while exhibiting low toxicity.
To clarify the mechanism of action, the authors investigated the binding interaction between resveratrol and the viral M protein using STD-NMR and fluorescence spectroscopy. These experiments confirmed the binding of resveratrol to the M protein, suggesting that the interaction is mainly driven by hydrophobic effect. In addition, they identified a tryptophan residue which interacts with the ligand.
Building on these findings, the authors used molecular docking and molecular dynamics simulations to predict the binding site and investigate the interactions. The simulations showed that binding site 1 is the most likely interaction site. The study further showed that the interaction between resveratrol and the M protein is stabilized by a combination of hydrogen bonding and hydrophobic interactions. This site, located at the dimerization interface of the M protein, includes a tryptophan residue (W242). The methods used were well performed and adequately described.
However, I have a few key comments:
The authors mention the determination of EC50 and selectivity index (SI) in their methods for the antiviral assays, but these data are missing in the results section. Only inhibition values for three concentrations are given. It would be advantageous to provide the complete concentration-response data for which the EC50 was determined.
Thank you for your suggestion. We have included an image illustrating the cellular viability of resveratrol alongside its antiviral activity (Figure 1).
This would give a clearer picture of the antiviral efficacy of the compound. In the Molecular Docking and Molecular Dynamics of M Protein-Resveratrol Interaction section of the results, the authors state that two different ligand conformations were observed at binding site 1-beta, but only one conformation is shown. This discrepancy needs to be addressed, and the manuscript should include visualizations or discussions of both conformations.
Thank you very much for your comment. We indeed mentioned the two conformations but did not include them in the text. We have added a new figure in the supplementary information (Figure S5) that shows both conformations and their interactions (from the most representative structure of each). This difference is captured by the RMSD coordinate (Figure 5c); however, it represents only a subtle change in the A-ring of resveratrol, as it remains at the same site and shares the same types of interactions with similar residues.
The first three paragraphs of the Discussion focus on the inflammatory response associated with RSV infection. This content seems unnecessary, since the study does not directly examine inflammation or immune responses.
As suggested, we removed the discussion regarding interleukins and modulators. We added a few lines to contextualize the role of resveratrol as an antiviral. Additionally, we retained the discussion of a study that evaluated the effects of resveratrol on RSV infection in rats, as we believe that studies in animal models supporting the efficacy of resveratrol strengthen our results.
The manuscript lacks a comprehensive discussion of the toxicity and antiviral activity of resveratrol. This would be an important addition to better contextualize the therapeutic potential of resveratrol.
Thank you very much for your suggestion. We have added information about the toxicity and antiviral activity of resveratrol to the main text, as well as a new figure that presents this information as results (lines 245 to 252 and Figure 1).
In line 290, the authors state that Molecular dynamics simulations demonstrated the stability of this bond through RMSD….; but no RMSD values are provided.
Thank you very much for your comment. We have added the values related to the RMSD graph shown in Figure 5, which indicate a slight variation in the values (0.2 nm) and excellent stability of the complex for both 1-alpha and 1-beta sites. Additionally, we have included a new image in the supplementary information that displays the ligand structure and its interactions, illustrating the two conformations accessed throughout the molecular dynamics that are responsible for the variation of the ligand's RMSD at the 1-beta site.

Round 2
Reviewer 1 Report
Comments and Suggestions for Authors
The answers to my questions got resolved
line 22: "over 48 hours,..." should be h
line 302: by Drª Karina Alves, what is it mean the a in superscript? I will remove it
Lines 335-336: "37ºC until it reached an absorbance of 0.6. The expression was 335 induced at 28oC..." use a single one type of degree symbol
Line 350: "50 mM Na2HPO4/NaH2PO4..." use the convention here and all the manuscript for chemical descriptors the 2 and 4 must be subscript as convention.
The authors explain well in the answers the source of resveratrol, but they did not include in the manuscript.
Also, I missed some reference that resveratrol also can be Cis and this can be formed in the body by itself.
The authors are invited to double check for the conventions in the manuscript other than that the manuscript is suitable for publication with minor changes
Author Response
Interaction of HRSV M protein with resveratrol shows antiviral effect
Thaina Rodrigues; Jefferson de Souza Busso; Raphael Vinicius Rodrigues Dias; Isabella Otenio Lourenço; Jessica Maróstica de Sa; Sidney Jurado de Carvalho; Icaro Putinhon Caruso; Fatima Pereira de Souza; Marcelo Andres Fossey*
Email: marcelo.fossey@unesp.br
Response letter to the reviewers.
The changes and suggestions that were added in this second answer are marked in green.
Reviwer 1
The answers to my questions got resolved
Line 22: "over 48 hours,..." should be h
Thank you for your comment. As suggested, we have modified this detail in the text.
Line 319: by Drª Karina Alves, what is it mean the a in superscript? I will remove it
Thank you for your comment and suggestion. We have taken it into account and removed it from the text.
Lines 335-336: "37ºC until it reached an absorbance of 0.6. The expression was 335 induced at 28oC..." use a single one type of degree symbol.
Thank you for pointing out this detail. As suggested, we have adjusted and standardized the symbol in the text.
Line 350: "50 mM Na2HPO4/NaH2PO4..." use the convention here and all the manuscript for chemical descriptors the 2 and 4 must be subscript as convention.
Thank you for pointing out this detail. As suggested, we have adjusted the chemical descriptors in the text.
The authors explain well in the answers the source of resveratrol, but they did not include in the manuscript.
Thank you for your comment. We have added the information regarding the molecule in section Materials and Methods 4.2 “Cytotoxicity Assay” the information on purity, origin, and isomeric trans.
Also, I missed some reference that resveratrol also can be Cis and this can be formed in the body by itself.
Thank you for your comment. As suggested, we have added this information on lines 59-62.
The authors are invited to double check for the conventions in the manuscript other than that the manuscript is suitable for publication with minor changes.
Thank you very much for all the comments and suggestions for our work. We have incorporated all of them and highlighted the modifications in the text.

Reviewer 2 Report
Comments and Suggestions for Authors
Nothing more
Author Response
Thank